# Identifying Stabilin-1 and Stabilin-2 Double Knockouts in Reproduction and Placentation: A Descriptive Study

**DOI:** 10.3390/ijms21197235

**Published:** 2020-09-30

**Authors:** Soon-Young Kim, Eun-Hye Lee, Eun Na Kim, Woo-Chan Son, Yeo Hyang Kim, Seung-Yoon Park, In-San Kim, Jung-Eun Kim

**Affiliations:** 1Department of Molecular Medicine, Cell and Matrix Research Institute, School of Medicine, Kyungpook National University, Daegu 41944, Korea; ksygood741@naver.com (S.-Y.K.); coramdeoeh@nate.com (E.-H.L.); 2BK21 Plus KNU Biomedical Convergence Program, Department of Biomedical Science, Kyungpook National University, Daegu 41944, Korea; 3Department of Pathology, Asan Medical Center, University of Ulsan College of Medicine, Seoul 05505, Korea; hk1997@naver.com (E.N.K.); wcson@amc.seoul.kr (W.-C.S.); 4Department of Pediatrics, School of Medicine, Kyungpook National University, Daegu 41944, Korea; kimyhmd@knu.ac.kr; 5Division of Pediatric Cardiology, Kyungpook National University Children’s Hospital, Daegu 41404, Korea; 6Department of Biochemistry, School of Medicine, Dongguk University, Gyeongju 38066, Korea; psyoon@dongguk.ac.kr; 7Center for Theragnosis, Biomedical Research Institute, Korea Institute of Science and Technology, Seoul 02792, Korea; iskim14@kist.re.kr; 8KU-KIST Graduate School of Converging Science and Technology, Korea University, Seoul 02841, Korea

**Keywords:** Stabilin-1, Stabilin-2, double knockout, placenta, decidua, hemorrhage

## Abstract

The placenta undergoes reconstruction at different times during fetal development to supply oxygen and nutrients required throughout pregnancy. To accommodate the rapid growth of the fetus, small spiral arteries undergo remodeling in the placenta. This remodeling includes apoptosis of endothelial cells that line spiral arteries, which are replaced by trophoblasts of fetal origin. Removal of dead cells is critical during this process. Stabilin-1 (Stab1) and stabilin-2 (Stab2) are important receptors expressed on scavenger cells that absorb and degrade apoptotic cells, and Stab1 is expressed in specific cells of the placenta. However, the role of Stab1 and Stab2 in placental development and maintenance remain unclear. In this study, we assessed Stab1 and Stab2 expression in the placenta and examined the reproductive capacity and placental development using a double-knockout mouse strain lacking both Stab1 and Stab2 (Stab1/2 dKO mice). Most pregnant Stab1/2 dKO female mice did not produce offspring and exhibited placental defects, including decidual hemorrhage and necrosis. Findings of this study offer the first description of the phenotypic characteristics of placentas and embryos of Stab1/2 dKO females during pregnancy, suggesting that Stab1 and Stab2 are involved in placental development and maintenance.

## 1. Introduction

Stabilin-1 (Stab1), belongs to a family of scavenger receptors, is a type 2 macrophage marker and is expressed in tissue macrophages and different endothelial cell subtypes [1,2,3]. Stabilin-2 (Stab2) belongs to the same family of scavenger receptors and shares 55% identity with Stab1 at the protein level [1,2,4,5]. Both Stab1 and Stab2 are expressed in macrophages and endothelial cells of various organs, including the liver and spleen [6,7]. Stab1 acts as a scavenger receptor for clearing debris and apoptotic cells through macrophage phagocytosis and is involved in the transcytosis of several components, including placental lactogen (PL) [1,3,8,9]. Stab1 modulates angiogenesis, supports leukocyte adhesion and transmigration, and regulates immune responses [1,3,9]. Stab2 is involved in clearing noxious blood factors and degrading apoptotic cells [10]. Stab2 also regulates extracellular matrix turnover and hyaluronan resorption to maintain body fluids, and participates in the defense against bacterial infections and recruitment of lymphocytes [6,7].

During pregnancy, the placenta undergoes structural changes to transfer maternal nutrients, oxygen, metabolites, and hormones to the fetus at different stages of development. These alterations are essential for the development and growth of the fetus [9,11,12]. One such change that occurs in early pregnancy is that the diameter of maternal blood vessels increases to supply a massive amount of maternal blood through the placenta at a low blood pressure. This phenomenon is called spiral artery remodeling, which is an essential change in the development of the placenta. During spiral artery remodeling, fetal origin trophoblasts invade maternal spiral arteries and replace endothelial cells undergoing apoptosis [12,13]. Apoptotic cells and debris are eliminated by placental macrophages and uterine natural killer (uNK) cells [12,13]. Rapid removal of apoptotic cells and debris is important for spiral artery remodeling. Inhibition of such removal hinders spiral blood vessel formation [12,13].

Stab1 and Stab2 proteins are expressed in endothelial cells and macrophages, which play a predominant role in artery remodeling in the placenta during pregnancy [12]. These proteins are closely related to factors mediating cell signaling pathways that regulate angiogenesis [4,9,14]. Functional studies of Stab1 and Stab2 in tissue macrophages have demonstrated their importance in the phagocytosis of apoptotic cells, suggesting that Stab1 and Stab2 contribute to placental reconstruction during pregnancy [1,2,3,10]. Interestingly, several studies have reported that Stab1 is expressed in placental endothelial cells and macrophages during pregnancy, and that it might be involved in cell adhesion and molecular scavenging in placental macrophages [1,3,9]. Previous studies have also suggested that Stab1 regulates differences in the concentration of mouse PL in maternal and fetal blood circulation [1,9]. However, the role of Stab1 in the placenta remains poorly understood and, to the best of our knowledge, the role of Stab2 in the placenta has not been reported yet.

In this study, we analyzed the mRNA expression levels of Stab1 and Stab2 in the placenta and compared them with those in the liver. In addition, we generated a double-knockout mouse strain lacking both Stab1 and Stab2 (Stab1/2 dKO mice) and observed defective reproduction in pregnant Stab1/2 dKO female mice. Furthermore, to determine the causes underlying these reproductive defects, we conducted a descriptive study to phenotypically characterize reproduction and placentation in pregnant Stab1/2 dKO female mice. This study provides the first description of Stab1/2 dKO affecting reproductive capacity and placental development, suggesting that Stab1 and Stab2 play an important role in placental reconstruction and remodeling during pregnancy.

## 2. Results

### 2.1. Stab1 and Stab2 mRNA Expression in Mouse Placenta

To determine whether Stab1 and Stab2 are expressed in the mouse placenta, mRNA expression levels of Stab1 and Stab2 in mouse placentas were measured using quantitative polymerase chain reaction (qPCR) and reverse transcription-PCR (RT-PCR) and compared with their expression levels in the liver as a positive control. Stab1 expression was 1.4-fold higher in the placenta than in the liver, whereas Stab2 expression in the placenta was half of the expression level in the liver, as determined using qPCR (Figure 1A) and confirmed using RT-PCR (Figure 1B).

### 2.2. Viability of Stab1/2 dKO Female Mice during Pregnancy

After intercrossing male and female Stab1/2 dKO mice to generate Stab1/2 dKO embryos (Figure 2), some pregnant Stab1/2 dKO females died during pregnancy without any physical sign of wounds, whereas other pregnant females abruptly lost their visible pregnancies before delivery. Mating between Stab1/2 dKO mice was performed at 7–8 weeks of age. The females were separated from the males into a different cage at week 2 after mating, and their pregnancy was confirmed by visual observation (swollen belly) every day. A total of seven Stab1/2 dKO females were used in the study and each of them showed 1–3 times visually identifiable pregnancies. Of these, 78% of visually identifiable pregnancies were lost without delivery, including loss due to maternal death. During pregnancy, some females exhibited abnormal behaviors and physical features, including reduced belly size, lethargic movements and responses to external stimuli, and premature delivery. To understand why pregnant Stab1/2 dKO female mice spontaneously died or lost their progeny, mice were sacrificed and dissected immediately when they exhibited these abnormal signs. Upon dissection of the skin and peritoneum, uteri of Stab1/2 dKO female mice were collected at embryonic day 13.5 (E13.5) and E17.5. Some portions of the uterine horns at E17.5 were filled with blood (Figure 3A). This phenomenon was also observed in embryos of uteri of Stab1/2 dKO pregnant mice at E13.5 (Appendix A). Moreover, the uteri of Stab1/2 dKO female had unevenly sized embryos or undefined tissue masses (Appendix A). Some pregnant Stab1/2 dKO female mice showed embryo resorptions; however, it was difficult to determine the number of embryos that were present before embryo resorption into the uterus. Instead, the number of viable implantation sites in the uteri of WT and Stab1/2 dKO female mice at E17.5 was determined (Appendix A). Chorioamniotic membranes surrounded undefined lumps, and bleeding was also observed. After dissecting these chorioamniotic membranes in pregnant mice with embryos, brown blood spreading and embryos were visible (Figure 3B–D). The shape of each placenta was maintained, but surfaces of some placentas were surrounded by decomposed membranes. The lower abdomen of each embryo was connected to the placenta through an umbilical cord that was degraded in some embryos (Figure 3C). Three of seven Stab1/2 dKO females used in this study died during the first delivery, indicating a maternal mortality rate of 42.86% at first delivery in Stab1/2 dKO female mice (Figure 3E). Stab1/2 dKO male mice had no abnormalities before or after mating and no death until the age of 50 weeks (n = 6, data not shown).

It was difficult to obtain a large number of Stab1/2 dKO mice required for this study because of spontaneous deaths of pregnant Stab1/2 dKO female mice or loss of their progeny. To determine the cause of these phenomena, male mice with genotypes of Stab1 heterozygous and Stab2 knockout were intercrossed with female mice carrying the same genotypes. Female mice bred well, and the total number of offspring per litter was 4–7 (data not shown). However, the number of offsprings with the Stab1/2 dKO genotype was extremely low compared with the number of pups with other genotypes (Appendix A). These results indicate that both Stab1 and Stab2 are important in the placenta and in the embryonic development during pregnancy as well as in the pregnant female mouse itself.

### 2.3. Phenotypic and Histological Defects of Placentas from Stab1/2 dKO Females

To characterize the phenotypic defects observed, the placentas and embryos from pregnant WT and Stab1/2 dKO mice were harvested at E17.5 (Figure 4A). Observation of E17.5 fetuses in the WT and Stab1/2 dKO uteri revealed that some Stab1/2 dKO fetuses appeared normal, while others were difficult to observe as they were either degraded or resorbed. Interestingly, some Stab1/2 dKO fetuses were normal in shape, although their placentas and other extraembryonic structures appeared defective with brownish discoloration (Figure 4B).

Histological analysis of placental tissues was performed to investigate external defects of placentas in Stab1/2 dKO mice. In both WT and Stab1/2 dKO mice, placentas had three layers: decidua, junctional zone, and labyrinth zone (Figure 5A). The length of each placental zone was quantified over the total length of the placenta expressed as a percentage. In Stab1/2 DKO placenta, the decidual length was increased relative to the total length of the placenta (Figure 5A). Glycogen trophoblast cells accumulate glycogen in the junctional zone of mouse placentas until E12.5, invade interstitially into the decidua until E18.5, and finally decrease in number [11,15]. A higher number of glycogen-containing clear cells was observed in the junctional zones of placentas obtained from Stab1/2 dKO mice than in those from WT mice based on hematoxylin and eosin (H&E) staining (Figure 5B). Periodic acid-Schiff (PAS) staining confirmed a higher number of cells containing glycogen in the junctional zones of placentas derived from Stab1/2 dKO mice than those from WT mice (Figure 6A,B). PAS-positive stained area per total tissue area was analyzed using ImageJ. Glycogen cells were significantly increased in the placenta of Stab1/2 dKO mice compared with that of WT mice (Figure 6C). Decidual hemorrhage, decidual necrosis, and fibrin deposition were also observed in the placental decidua of Stab1/2 dKO mice (Figure 5A and Figure 7). Deciduae of placentas derived from Stab1/2 dKO mice were thicker than those of WT mice due to decidual and retroplacental hemorrhage.

## 3. Discussion

This study provides a phenotype description of reproduction, placentation, and embryo development in Stab1/2 dKO female mice. The phenotype observed in these mice was similar to human placental abruption. In pregnant mice lacking both Stab1 and Stab2, the placental decidua layer was expanded due to decidual hemorrhage and necrosis. During pregnancy, the separation of the placenta and endometrial walls results in placental abruption and the loss of nutrient and oxygen supply from the mother, leading to miscarriage [16,17]. Decidual bleeding occurs due to abnormalities in the early development of spiral arteries, which is one of the causes of placental destruction [16,17]. Spiral artery remodeling occurs in the decidua, which is comprised of specialized endometrial stromal cells to sustain the pregnancy. The decidua is made of maternal blood vessels as well as numerous leukocytes such as decidual macrophages and uNK cells known to contribute to artery remodeling [12,13,18,19,20]. Artery remodeling occurs through a specific mechanism in which trophoblasts replace endothelial cells of arteries and blood vessels in transit to spiral arteries [12,13,18]. Formation of spiral arteries and replacement of trophoblasts depends on the apoptosis of endothelial cells and degradation of the extracellular matrix by decidual cells [12,13,18]. Decidual macrophages and uNK cells produce various matrix metalloproteinases, angiogenic factors, and cytokines that induce endothelial cell apoptosis [9,12,19]. These cells scavenge residues due to spiral artery formation and modulate immune responses in the placenta [12,13,18,19,20]. Stab1 and Stab2 participate in the elimination of apoptotic cells via macrophage phagocytosis [1,2,4,5]. Stab1 and Stab2 are involved in the regulation of the extracellular environment and modulation of angiogenesis [1,4,9,14]. Stab2 also plays a role in maintaining fluidity of the blood that passes through arteries [7]. Schledzewski et al. (2011) previously investigated Stab1/2 dKO mice, which were generated by crossing Stab1 KO mice with Stab2 KO mice [10]. These mouse lines were generated using different strategies for targeting Stab1 and Stab2, and thus differ from the ones used in this study [8,21]. Their Stab1/2 dKO mice exhibited significantly lower survival rate, impaired liver blood clearance capacity, and increased kidney fibrosis compared with WT mice [10]. However, their study did not focus on elucidating the cause of the low survival rate in Stab1/2 dKO mice. Moreover, no specific physical feature was reported in each Stab1 KO and Stab2 KO mouse line [8,21] used in the present study. Here, Stab1/2 dKO progeny obtained from Stab1/2 dKO female mice were fewer than the progeny derived from WT female mice. As it was difficult to obtain a sufficient number of Stab1/2 dKO mice, no further studies were conducted to investigate the detailed molecular mechanisms underlying Stab1/2-deficient placental defects. Moreover, due to increased body waste that occurs in pregnancy, kidney function is important and metabolic imbalances such as gestational diabetes might develop in some cases [22]. Thus, it is necessary to understand the role of Stab1 and Stab2 in kidney function during pregnancy. Here, we present a possibility that Stab1 and Stab2 are associated with deciduous hemorrhage in the placenta. These findings suggest that Stab1 and Stab2 deficiency leads to a failure in clearing apoptotic cells following arterial remodeling, thereby affecting the angiogenic function of macrophages and blood fluidity. Such failure eventually results in placental defects. Additional studies are warranted to determine how Stab1 and Stab2 deficiency induces reproduction defects and to further elucidate the series of events leading to placental hemorrhage due to Stab1 and Stab2 deficiency. Our results suggest that decidual hemorrhage due to Stab1 and Stab2 deficiency can lead to serious defects in the placenta as well as reproduction failure.

The junctional zone of the placenta contains glycogen cells that accumulate glycogen during pregnancy [11]. These cells migrate into the maternal decidua after E12.5, and act as an energy reservoir for the mother and/or fetus during late-stage gestation in mice [11,13]. The number of glycogen cells declines by half by E18.5 [11]. The placentas at E17.5 carried a higher number of glycogen cells in the Stab1/2 dKO female mice than in WT mice, suggesting that the lack of Stab1 and Stab2 in the placenta hinders the accumulation and use of glycogen. Issues associated with glycogen accumulation in the placenta often occur due to inhibition of trophoblast invasion and spiral artery remodeling or due to suppression of vascular formation and expansion [11]. In particular, blood vessels and conduits are associated with abnormal formation of junctional zones [23]. Defects in the junctional zone can alter the endocrine environment in the placenta, resulting in systemic effects on both the fetus and the mother [23]. The role of Stab1 and Stab2 in the regulation of glycogen accumulation and use in the placenta remains unclear and is difficult to define. However, histological data suggest that the lack of Stab1 and Stab2 leads to defects in glycogen metabolism in mouse placenta. In addition, Stab1 and Stab2 deficiency may lead to alterations in placental morphology and function due to abnormal regulation of glycogen, at least in part. The junctional zone is an endocrine zone that secretes several pregnancy hormones [13]. Stab1 is involved in the transcytosis of placental hormones, including PL [1,9]. PL, which has a similar structure and function as the growth hormone, regulates lactogenesis, maintenance of the uterine wall, and the synthesis of progesterone during late pregnancy [1,9]. Moreover, PL induces lipolysis, amino acid mobilization, and glycogenolysis to facilitate nutrient delivery to the fetus and promotes glycogen accumulation in the junctional zone by glycogen trophoblasts. Interestingly, the concentration of mouse PL differs between maternal (10 µg/mL) and fetal (0.5 µg/mL) circulation [9]. This difference occurs due to the lack of direct connection between maternal and fetal circulatory systems. Placental macrophages are present in these circulatory systems, and PL concentration is controlled by the transcytosis of placental macrophages. Stab1 is a receptor involved in PL transcytosis of placental macrophages [9]. Stab1 deficiency in the placenta can disrupt the regulation of PL concentrations in maternal and fetal circulation, ultimately leading to placental abnormalities [9]. Moreover, Stab1 and Stab2 are involved in cell migration and extracellular matrix regulation, suggesting that they contribute to the proper migration of placental cells during pregnancy [9]. The present study also suggests that Stab1 and Stab2 are important to regulate the function of the junctional zone.

This study has a limitation that it could not conclude the role of Stab1 and Stab2 in placental development because it was difficult to obtain sufficient Stab1/2 dKO mice. Further studies are needed to ascertain that the placental changes observed in Stab1/2 dKO females are due to the lack of Stab1 and Stab 2 expression using a placental-specific Stab1/2 knockout mouse model. Furthermore, embryo transfer studies transplanting Stab1/2 dKO blastocysts into WT surrogates will be used to better determine the role of Stab1 and Stab2 in mammalian placentation. Moreover, additional work is needed to determine the detailed molecular mechanism(s) underlying this process. In conclusion, we observed that Stab1/2 dKO female mice exhibited reproduction failure owing to placental defects, including hemorrhage in the decidua and abnormalities in the junctional zone. These findings provide the first description of the reproductive capacity and phenotypic characteristics of placenta and embryos due to Stab1 and Stab2 deficiency in mice.

## 4. Materials and Methods

### 4.1. RNA Extraction, RT-PCR, and qPCR

Placentas and livers were isolated from WT pregnant mice at E17.5. The samples were immersed in TRIzol reagent (Ambion, Austin, TX, USA) and lysed to extract RNA using Qiagen tissue lyser II, as previously described [5]. RNA was then reverse transcribed into cDNA using a Reverse Transcription Master Premix (Elpis Biotech, Daejeon, Korea) for RT-PCR and qPCR. RT-PCR was performed at an annealing temperature of 64 °C for Stab1 and 60 °C for Stab2 (35 cycles), and 60 °C for Gapdh as a loading control (25 cycles). Results of RT-PCR were analyzed using ImageJ (National Institute of Health, Bethesda, MD, USA). qPCR was performed using Applied Biosystems StepOnePlus qPCR system (Life technologies Software v2.3; Life technologies, Carlsbad, CA, USA) under the following conditions: initial denaturation at 95 °C for 10 min, 40 cycles of denaturation at 95 °C for 15 s and amplification at 60 °C for 1 min, and a final cooling step. Results of qPCR were analyzed using the comparative cycle threshold (*C_T_*) method. Primer sets used for this experiment are as follows: *Stab1*, 5′-TGC GAC ATC CAC ACC AAG TT-3′ and 5′-TGA ACC ACA TCC TTC CAG CA-3′; *Stab2*, 5′-AGC TGC TGC CTT TAA TCC TCA-3′ and 5′-ACT CCG TCT TGA TGG TTA GAG TA-3′; and *Gapdh*, 5′-GCA TCT CCC TCA CAA TTT CCA-3′ and 5′-GTG CAG CGA ACT TTA TTG ATG G-3′.

### 4.2. Animals and Sampling

Stab1/2 dKO mice were generated by crossing Stab1 KO mice with Stab2 KO mice [8,21]. Male and female Stab1/2 dKO mice were intercrossed to generate Stab1/2 dKO embryos and placentas (Figure 2). The mating of males and females for pregnancy was performed at the ratio of 1:1 in a single cage; and 2 weeks after mating, the females were separated into different cages and followed until birth or pregnancy loss (observation for at least 20 days after separation from the males). The date of pregnancy of the pregnant females was determined by the normal gross fetal anatomy of embryos obtained in their uteri and the comparison based on the text “The atlas of mouse development (edited by M. H. Kaufman, Elsevier academic press)”. The genotypes of mice and/or embryos were analyzed using PCR and genomic DNA isolated from tails or yolk sacs. Primer sets and PCR amplification conditions for Stab1 and Stab2 were previously described [8,21]. WT females or embryos were used as controls. The mice were bred and maintained in climate-controlled (20–25 °C), specific pathogen-free conditions with a 12 h/12 h light/dark cycle and were allowed free access to standard mouse diet and water. All procedures concerning animal experiments were conducted with the approval of the Institutional Animal Care and Use Committee of Kyungpook National University (Approval number KNU-2016-0011, Daegu, Korea).

### 4.3. Histological Analysis

Embryos and placentas were isolated from pregnant female mice at E17.5. Placentas were fixed overnight in 4% paraformaldehyde at 4 °C, washed in distilled water, dehydrated through an ethanol series, embedded in paraffin, and sectioned to 3-μm thickness using a microtome (Leica Biosystems, Wetzlar, Germany). These sections were stained with H&E and PAS. For PAS staining, after deparaffinization and rehydration, sections were dipped in 0.5% periodic acid (Sigma-Aldrich, St. Louis, MO, USA) for 5 min and then treated with Schiff’s reagent (Sigma-Aldrich) for 15 min. The stained sections were observed under a microscope slide scanner (Microscope Central, Feasterville, PA, USA), and PAS-positive stained area per total tissue area was analyzed by using ImageJ (NIH, Bethesda, MD, USA).

### 4.4. Statistical Analysis

Data were analyzed by Student’s *t*-test and presented as means ± standard deviation (SD). A *p* value < 0.05 was considered statistically significant.

## Figures and Tables

**Figure 1 ijms-21-07235-f001:**
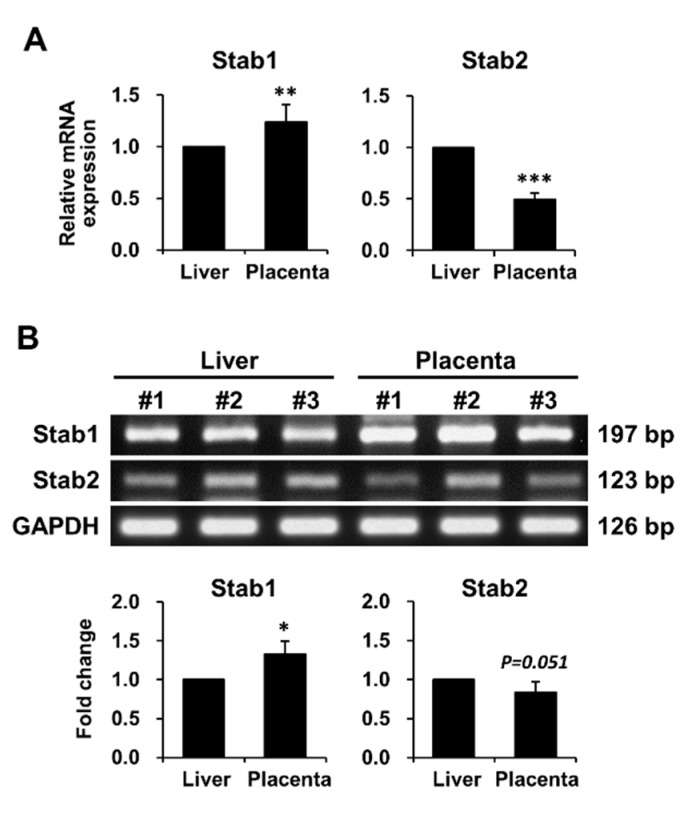
mRNA expression levels of Stab1 and Stab2 in WT female mouse placentas. The mRNA expression levels of Stab1 and Stab2 in mouse placentas were measured using qPCR (**A**) and RT-PCR (**B**) and were compared with those in the liver used as a positive control. The relative expression levels shown in (**A**) were plotted against those of the respective genes in the liver, which were set to 1.0. Intensities of individual bands obtained via RT-PCR shown in (**B**) were determined using ImageJ. Data were normalized to Gapdh expression and calculated as fold change relative to the expression level in the liver, which was set to 1.0. Numbers in (**B**) indicate the representative mice used in the experiment. *, *p* < 0.05; **, *p* < 0.01; ***, *p* < 0.001 versus the positive control; n = 3.

**Figure 2 ijms-21-07235-f002:**
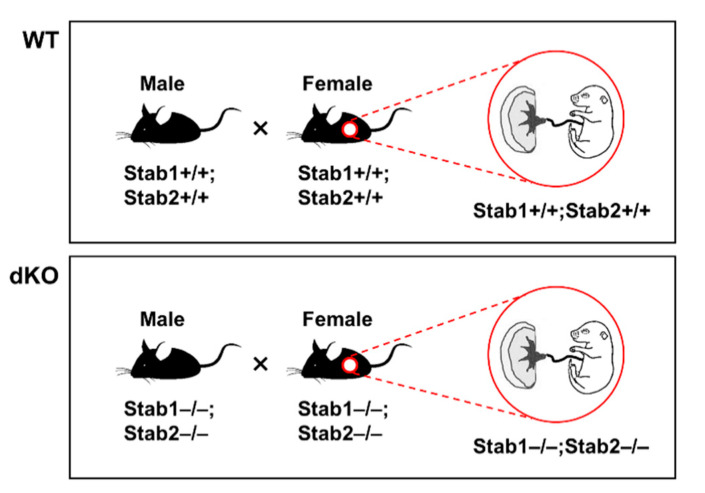
Pedigree and genotypes for obtaining embryos and placentas from pregnant females for the experiments.

**Figure 3 ijms-21-07235-f003:**
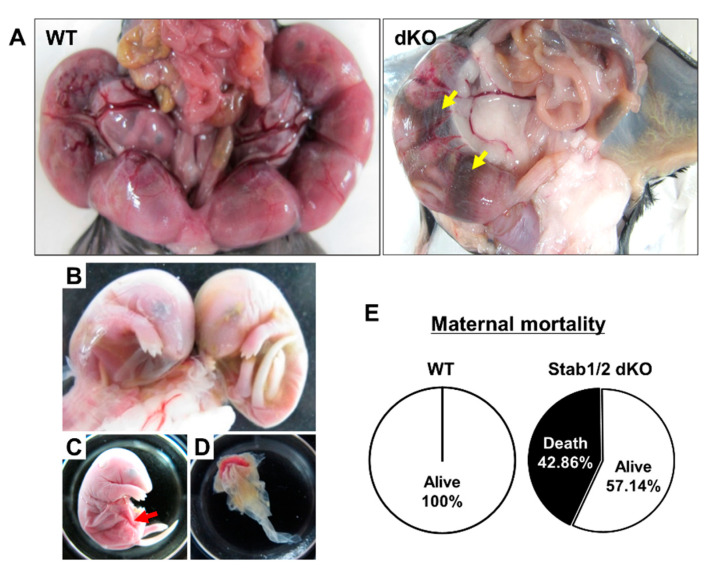
Embryos from pregnant Stab1/2 dKO mice at E17.5. (**A**) Uterine horns with embryos in WT and Stab1/2 dKO mice. Yellow arrows in Stab1/2 dKO indicate blood within the uterine horns. (**B**) Embryos isolated from the uterine horns of a Stab1/2 dKO female mouse. (**C**) Embryo and (**D**) placenta from a Stab1/2 dKO female mouse. The fetus only maintained the outer shape in the lower body and the fetal interior was degraded near the umbilical cord. Red arrow in (**C**) denotes abnormal abdomen in an embryo. (**E**) Maternal mortality of Stab1/2 dKO mice at the first delivery. n = 7.

**Figure 4 ijms-21-07235-f004:**
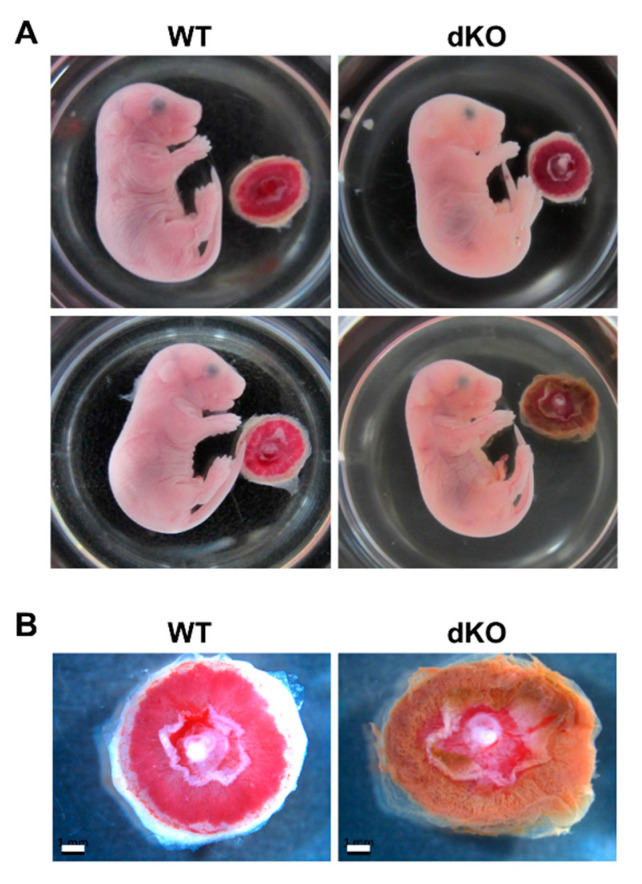
Embryo and placenta from pregnant WT and Stab1/2 dKO mice. (**A**) WT and Stab1/2 dKO embryos at E17.5 and associated placentas. (**B**) High-magnification images of the placentas derived from WT and Stab1/2 dKO mice. Scale bar, 1 mm; n = 3.

**Figure 5 ijms-21-07235-f005:**
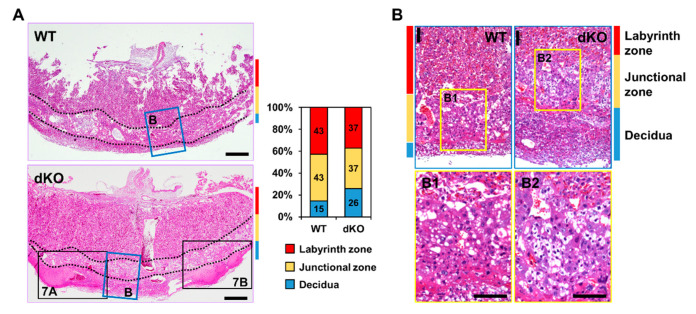
Histological analysis of WT and Stab1/2 dKO placentas. (**A**) H&E staining showing the overall shape and structure of placentas from WT and Stab1/2 dKO mice. Three layers of the placenta are indicated by colored lines, with red indicating the labyrinth zone, yellow indicating the junctional zone, and blue indicating the decidua. Quantitative analysis of the length of each placental zone. The percentages indicated for decidua, junctional zone, and labyrinth zone were quantified over the total length (100%) of the placenta. Scale bar, 500 µm. (**B**) High-magnification images of the blue rectangles in (**A**). (B1, B2) High-magnification images of the junctional zone in (**B**). Scale bar, 100 µm.

**Figure 6 ijms-21-07235-f006:**
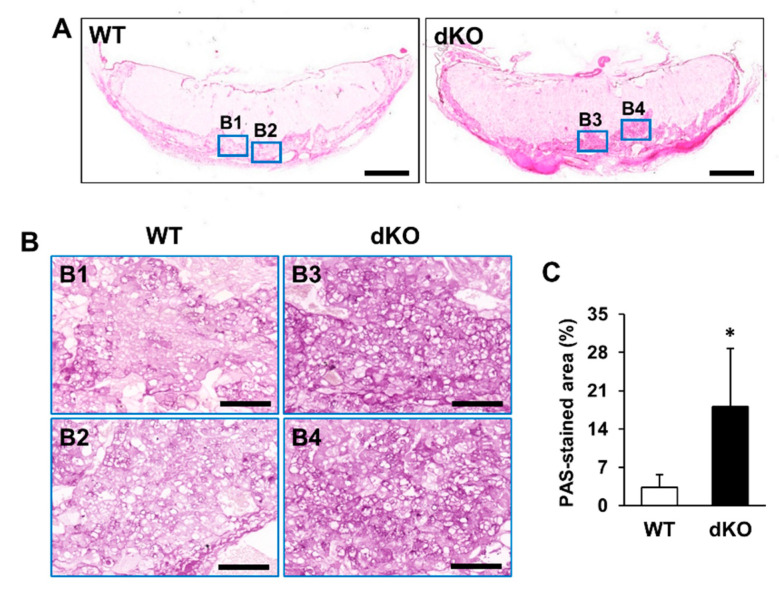
Histological analysis of WT and Stab1/2 dKO placentas based on PAS staining. (**A**) Overall shape of the PAS-stained placentas derived from WT and Stab1/2 dKO mice. Scale bar, 1 mm. (**B1**–**B4**) High-magnification images of the blue rectangles in (**A**). Scale bar, 100 µm. (**C**) Percentage of PAS-stained positive area to the total tissue area was analyzed using ImageJ. *, *p* < 0.05 versus WT; n = 3.

**Figure 7 ijms-21-07235-f007:**
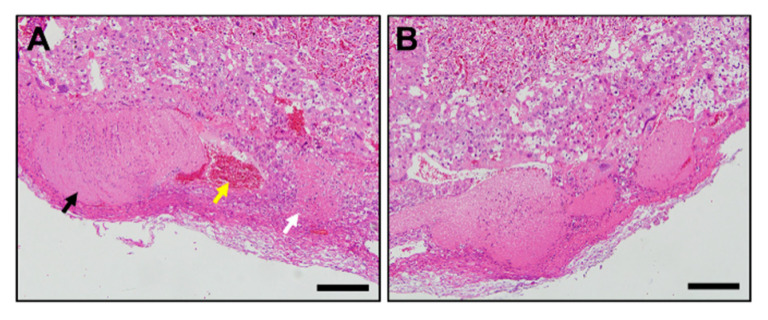
Histological analysis of the decidua of Stab1/2 dKO placenta. (**A**,**B**) High-magnification images of the black rectangles in the placenta from Stab1/2 dKO mouse shown in Figure 5A. In (**A**), black, yellow, and white arrows indicate decidual hemorrhage, maternal artery, and decidual necrosis with fibrin deposition, respectively. Scale bar, 200 µm.

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
