# Peer review of "Identifying Stabilin-1 and Stabilin-2 Double Knockouts in Reproduction and Placentation: A Descriptive Study"

_ijms, 2020, doi:10.3390/ijms21197235_

Round 1

Reviewer 1 Report

The current paper is based on knockout mice for Stab1 and Stab2 produced in previous papers by the authors. And basically in this paper the authors are presenting the breeding potential of Stab1/2 knockouts, and they tried to explore the link with placentation. In its current format the paper is quite descriptive rather than experimental as such. Although the information presented is novel, in its current format it is not clear if the findings were actually induced by Stab1/2 knockout or were just present by chance. In other words, the authors need to provide clear evidence that the phenotypes present in knockouts were not present (or in less proportion) in the WT counterparts. Hence, a more detailed statistical analysis is needed. Please address the following points:

1) The authors need to provide detail information about the conditions in which the animals (for both knockouts and WT animals) were kept during the whole experimental period (e.g. light cycle, feeding regime, how pregnant animals were kept, etc.).

2) Please indicate the number of pregnant Stab1/2 knockout females that died (i.e. dead/total) and provide comparisons with the WT group, this is, was the mortality rate between knockouts and WT statistically significant? Also indicate more precisely at what day of pregnancy these pregnant animals died. Using the term “late stage of pregnancy” is too vague. In line 101 the authors mentioned that “….whereas other pregnant females abruptly lost their visible pregnancies before delivery”. Here the authors need to indicate how they monitored (e.g. cameras, visual observation, how often, etc.) and determined pregnancy loss in the animals used in the study. The latter is especially relevant as the authors indicated in lines 102-103 that “…mice were dissected immediately after either event”. If it was done “immediately”, how the authors determined that a pregnancy was not viable anymore? And please provide information on the day of pregnancy (i.e. range) that foetal loss was detected. Again, “before delivery” is too vague.

3) The authors mentioned that ”..uteri of Stab1/2 dKO female mice at embryonic day 17.5 (E17.5) had uneven sized embryos or an undefined tissue mass”. The authors need to indicate how conceptus/foetal size was determined and provide statistical analysis for these data (i.e. knockouts vs WT, please provide n=).

4) It was mentioned that some portions of uterine horns were filled with blood and that the uterine artery was thinner in knockouts. This is a very interesting finding, but it really needs to be substantiated with statistics (i.e. knockouts vs WT, please provide n=).

5) The authors need to indicate how embryo resorption was determined and provide statistical data (i.e. knockouts vs WT).

6) All the morphological and histological features of the placenta describe in the paper need to be analysed with statistics (i.e. knockouts vs WT, please provide n=). Please provide information regarding the image analysis of histological samples, this is, the methodology to count cells (e.g. “glycogen-containing clear cells”).

7) It will be good if you could compare the mortality rate observed in pregnant Stab1/2 dKO females with the mortality rate in non-pregnant Stab1/2 dKO females. This will help clarify whether pregnancy per se can increase mortality in these knockouts.

Author Response

Thank you for the review of our manuscript ijms-921937, newly titled Double knockout of stabilin-1 and stabilin-2 induces placental insufficiency and hemorrhage (The previous title was changed by the Reviewer #2 comment). We have performed the corrections/modifications in response to the comments raised by the Editor and the Reviewers. The point-by-point responses with these comments are indicated hereafter in blue. Changes are yellow-highlighted through the revised manuscript.

****************************************************************

Reviewer #1

The current paper is based on knockout mice for Stab1 and Stab2 produced in previous papers by the authors. And basically in this paper the authors are presenting the breeding potential of Stab1/2 knockouts, and they tried to explore the link with placentation. In its current format the paper is quite descriptive rather than experimental as such. Although the information presented is novel, in its current format it is not clear if the findings were actually induced by Stab1/2 knockout or were just present by chance. In other words, the authors need to provide clear evidence that the phenotypes present in knockouts were not present (or in less proportion) in the WT counterparts. Hence, a more detailed statistical analysis is needed. Please address the following points:

1) The authors need to provide detail information about the conditions in which the animals (for both knockouts and WT animals) were kept during the whole experimental period (e.g. light cycle, feeding regime, how pregnant animals were kept, etc.).

>> Thank you for this comment. We have provided a detailed description regarding the conditions for animal breeding in the Results section (Lines 101-104) and in the Materials and methods section (Lines 289-291, 297-299). Briefly, the mice were bred and maintained in climate-controlled (20°C-25°C), specific pathogen-free conditions under a 12 h/12 h light/dark cycle and were allowed free access to a standard mouse diet and water. The first mating of males and females to induce pregnancy was performed at 7-8 weeks at a ratio of 1:1 in a single cage. During the second week after mating, the females were separated into a different cage and followed up until general delivery. The belly of the pregnant female was visually and everyday observed until at least 20 days after separation from the male.

2) Please indicate the number of pregnant Stab1/2 knockout females that died (i.e. dead/total) and provide comparisons with the WT group, this is, was the mortality rate between knockouts and WT statistically significant? 

>> Although it was not easy to obtain Stab1/2 dKO mice, 7 Stab1/2 dKO female mice were used in the study. During the experiment, a total of 14 pregnancies were observed in 7 Stab1/2 dKO females. Fourteen days after mating, the females were separated from the males and their bellies were visually and everyday observed for pregnancy for at least 20 days. A loss of pregnancy or maternal death before delivery occurred in 11 out of 14 pregnancies (78%). Pregnancy loss was determined by observing that the maternal pregnant belly became slim again without delivery and that the pregnant females did not give birth within 20 days after being separated from the males. Moreover, at first delivery, 3 out of 7 females were dead, indicating a maternal mortality of 42.86%. We have described the details in the Results section (Line 104-106, 123-125) and added the maternal mortality to Fig. 3E.

Also indicate more precisely at what day of pregnancy these pregnant animals died. Using the term “late stage of pregnancy” is too vague.

>> Thank you very much for the reviewer’s comment. To observe embryos and placenta, three pregnant Stab1/2 dKO females were sacrificed before death. During the visual observation, we found that pregnant females were struggling with abnormal signs: i.e. One female mouse exhibited evidence of pregnancy loss (the pregnant belly became slim without birth), hair defects, and lethargic movements and responses to external stimuli. The other female mouse was dying because of premature birth, with the offspring stuck in the vagina. Laparotomy was immediately performed when these abnormal signs were observed during pregnancy. To decide the day of pregnancy, we examined the gross fetal anatomy of embryos in uteri, based on the text “The atlas of mouse development (edited by M.H. Kaufman, Elsevier academic press)”. We described this in the Materials and methods section (Line 291-294) and have now specified the exact day of pregnancy by the embryos. We also revised the term “late stage of pregnancy” to “pregnancy” in the Results 2.2 section.

In line 101 the authors mentioned that “….whereas other pregnant females abruptly lost their visible pregnancies before delivery”. Here the authors need to indicate how they monitored (e.g. cameras, visual observation, how often, etc.) and determined pregnancy loss in the animals used in the study. The latter is especially relevant as the authors indicated in lines 102-103 that “…mice were dissected immediately after either event”. If it was done “immediately”, how the authors determined that a pregnancy was not viable anymore? And please provide information on the day of pregnancy (i.e. range) that foetal loss was detected. Again, “before delivery” is too vague.

>> Females were separated from the males after 14 days starting from the mating date, and their bellies were visually examined for evidence of pregnancy every day for at least 20 days after separation from the males. Pregnancy was determined by confirming the presence of an enlarged belly through visual observation, while the loss of pregnancy was identified as a maternal pregnant belly that became slim again in the absence of delivery during the 20 days after separation from the male. Unfortunately, we did not record images of the physical appearance of the pregnant females. As indicated in the response to the previous comment, the maternal mortality of Stab1/2 dKO females at first delivery was extremely high to obtain live pups. We observed the death of Stab1/2 dKO females several times during visually identifiable pregnancies. During pregnancy, some females exhibited abnormal behavior and physical features, including reduced belly size, slow movements and responses to external stimuli, and premature delivery. Therefore, we decided to sacrifice the pregnant females before their deaths and dissected them immediately after they began to display these abnormal signs. The day of pregnancy was determined by the normal gross fetal anatomy of the embryos in uteri, which was verified based on the text “The atlas of mouse development (edited by M.H. Kaufman, Elsevier academic press)”. This information has now been included in the Results 2.2 section (Line 106-110) and the Materials and methods section (Line 291-294). In this study, we observed that unevenly sized embryos from uteri or blood within uterine horns of Stab1/2 dKO females at E13.5 (Supplementary Fig. S1), but we were, unfortunately, unable to provide information on the exact date of pregnancy in which fetal loss was started because of the difficulty of obtaining and high mortality rate during pregnancy of Stab1/2 dKO mice.

3) The authors mentioned that ”..uteri of Stab1/2 dKO female mice at embryonic day 17.5 (E17.5) had uneven sized embryos or an undefined tissue mass”. The authors need to indicate how conceptus/foetal size was determined and provide statistical analysis for these data (i.e. knockouts vs WT, please provide n=).

>> We thank the reviewer for raising this point. As indicated in the response to the previous comment, we could not statistically determine the resorption rate or the exact number of unevenly sized embryos and undefined tissue masses in the uteri. Instead, we could count and compare the number of viable implantation sites in the uteri of WT and Stab1/2 dKO female mice at E17.5. We described this result in the Results section (Line 115-118) and Supplementary Fig. S3. In addition, the size and date of normal embryos were determined based on the text “The atlas of mouse development (edited by M.H. Kaufman, Elsevier academic press)”. To improve the readers’ understanding, we have added images of the unevenly sized embryos and the undefined tissue masses that were observed in the uteri of Stab1/2 dKO females in Supplementary Figs. S2 and S3.

4) It was mentioned that some portions of uterine horns were filled with blood and that the uterine artery was thinner in knockouts. This is a very interesting finding, but it really needs to be substantiated with statistics (i.e. knockouts vs WT, please provide n=).

>> From visual observation, we determined that some regions of the uterine horns were filled with blood and took pictures of this phenotype in Stab1/2 dKO females. Unfortunately, we did not measure the volume of blood or count the number of blood sites in the uterine horns of Stab1/2 dKO females compared with WT females. Instead, we have added images to show examples of uterine horns filled with blood in Stab1/2 dKO female mice at E13.5 (Supplementary Fig. S1) and E17.5 (Fig. 3). On the other hand, uterine horns were filled with embryos (no blood sites) in WT females at E17.5 (Supplementary Fig. S3). Regarding the thickness of the uterine artery, we observed that the uterine arteries were thinner in Stab1/2 dKO females than that in WT females, which were shown in Fig. 3. However, the thickness of the uterine artery was not measured for this study. We are very sorry not to be able to do statistical analysis, and we have decided to exclude this finding from the revised manuscript.

5) The authors need to indicate how embryo resorption was determined and provide statistical data (i.e. knockouts vs WT).

>> As indicated in the response to the previous comments, during the gestation period of pregnant Stab1/2 dKO female mice, we observed that the maternal pregnant belly was reduced in size without delivery or that maternal death occurred during a premature birth. We simultaneously observed the embryos and placentas from the females. Some embryos were resorbed, while other embryos turned into a lump of undefined tissue without revealing the shape of embryo. To support this result, we have provided a description and images of the embryos and others obtained from Stab1/2 dKO female mice (Supplementary Figs. S2 and S3). However, we did not quantify the number of embryos that existed before embryo resorption occurred; hence, we cannot provide statistical data. Instead, we have described this in the Results section (Line 115-118) and provided the number of viable implantation sites in the uteri of WT and Stab1/2 dKO female mice at E17.5 (Supplementary Fig. S3).

6) All the morphological and histological features of the placenta describe in the paper need to be analysed with statistics (i.e. knockouts vs WT, please provide n=). Please provide information regarding the image analysis of histological samples, this is, the methodology to count cells (e.g. “glycogen-containing clear cells”).

>> We appreciate and agree with the reviewer’s point. In response to the reviewer’s comment and to clarify the morphological and histological features of the placenta, we have now provided statistical data for some of our findings. Maternal mortality and the length of each placental zone were quantified in Fig. 3E and 5A, respectively (Line 123-125, Line 159-160). Glycogen-containing clear cells were stained with PAS (Fig. 6A,B), and we quantified the PAS-positive stained areas per total tissue area using ImageJ. Glycogen cells were significantly increased in the placentas of Stab1/2 dKO mice (p value = 0.038). We have described the details of this analysis in the Results section (Line 167-169), in addition to the statistical data presented in Fig. 6C, and the Materials and methods section (Line 309-310).

7) It will be good if you could compare the mortality rate observed in pregnant Stab1/2 dKO females with the mortality rate in non-pregnant Stab1/2 dKO females. This will help clarify whether pregnancy per se can increase mortality in these knockouts.

>> We appreciate and completely agree with the reviewer’s point. Stab1/2 dKO mice were generated by crossing Stab1 KO mice with Stab2 KO mice. In addition, Stab1/2 dKO mice were generated by crossing Stab1/2 dKO male mice with Stab1 Het;Stab2 KO female mice. However, it was difficult to obtain a large enough number of Stab1/2 dKO mice for experiments using either method. The comparison of mortality rates between non-pregnant and pregnant Stab1/2 dKO females required tracking of Stab1/2 dKO mice without mating. However, a sufficient number of Stab1/2 dKO mice could not be obtained, as mentioned. Interestingly, although not used in this experiment, one non-pregnant Stab1/2 dKO female mouse survived for > 50 weeks. However, we cannot insist that this mouse was representative of non-pregnant Stab1/2 dKO females. Therefore, we just showed the mortality rates of pregnant Stab1/2 dKO females at first delivery compared to WT females in Fig. 3E. In the case of Stab1/2 dKO male mice, they showed no abnormalities before or after mating and no death occurred until the age of 50 weeks (n = 6). The mortality rate of Stab1/2 dKO male mice was not focus on this study, but we have briefly described in the Result section (Line 125-126).

****************************************************************

The authors want to extend our sincere thanks to the Editor and the Reviewers for the helpful comments and the time they invested in reviewing our paper. We hope that this revision will satisfy the comments and requests from the Editor and the Reviewers as well as improve the overall quality of this manuscript. We also hope that the revised manuscript is now acceptable for publication in International Journal of Molecular Sciences.

Sincerely yours,

Jung-Eun Kim, Ph.D.

Department of Molecular Medicine

Cell and Matrix Research Institute

School of Medicine, Kyungpook National University

680 Gukchaebosang-ro, Jung-gu,

Daegu, 41944, Korea

Tel: 82-53-420-4949

Fax: 82-53-426-4944

Reviewer 2 Report

This is an interesting study describing the placental defects in crosses between mice with null mutations of Stab1 and Stab 2 genes. While the depth of analysis is somewhat preliminary, as noted by the authors, I consider that the significant phenotype observed is worth reporting as it will stimulate further study. Specific comments are listed below: 

Title:

Specify key finding in title. Double mutant of xxx causes xxx...

Abstract:

Line 27 clauses are reversed: ‘includes apoptosis of spiral arteries that line endothelial cells’

Introduction:

Grammar is somewhat unusual and some sentences require rewriting.

Lines 70-71: specify whether refers to mouse or human PL.

There is no significant description of the Stab 1 and Stab2 knockout mice, although references are given. A brief statement of their general non-placental phenotypes should be included as the maternal phenotype might affect placental phenotype in the current study.

Results:

Line 84 and elsewhere: ‘real-time PCR and RT-PCR’ are misleading. Real-time PCR is also ‘RT-PCR’. Use ‘quantitative real-time RT-PCR’ and ‘semi-quantitative RT-PCR’ or other more precise alternatives.

Lines 112 -113 and reference to ‘rotten eggs’ should be deleted.

Section 2.3. e.g., lines 143-144; Fig. 3; Fig. 4 etc: It is sometimes difficult to determine whether the stated genotypes refer to embryo or mother. There needs to be a more explicit specification of the maternal and embryonic genotypes for all experiments. It is unclear which effects are due to the maternal genotype and which due to the embryonic genotype in some experiments. The Supplementary Table does not clarify this issues although the explicit statement of numbers is welcome. Precise numbers should be stated throughout the Results section including in Figure legends.

Line 154: remove reference to human placental abruption / relocate to Discussion.

Discussion:

It would be good to see PL data in KO crosses, although the stated small numbers available may not have allowed this.

Author Response

Thank you for the review of our manuscript ijms-921937, newly titled Double knockout of stabilin-1 and stabilin-2 induces placental insufficiency and hemorrhage (The previous title was changed by the Reviewer #2 comment). We have performed the corrections/modifications in response to the comments raised by the Editor and the Reviewers. The point-by-point responses with these comments are indicated hereafter in blue. Changes are yellow-highlighted through the revised manuscript.

****************************************************************

Reviewer #2

This is an interesting study describing the placental defects in crosses between mice with null mutations of Stab1 and Stab 2 genes. While the depth of analysis is somewhat preliminary, as noted by the authors, I consider that the significant phenotype observed is worth reporting as it will stimulate further study. Specific comments are listed below: 

Title:

Specify key finding in title. Double mutant of xxx causes xxx...

>> Thank you very much for this comment. We have carefully modified the title to include the key finding. The revised title is “Double knockout of stabilin-1 and stabilin-2 induces placental insufficiency and hemorrhage”.

Abstract:

Line 27 clauses are reversed: ‘includes apoptosis of spiral arteries that line endothelial cells’

>> We appreciate the reviewer’s suggestion. We apologize for this error and have corrected the indicated sentence in Line 27.

Introduction:

Grammar is somewhat unusual and some sentences require rewriting.

>> A professional scientific editing service reviewed the manuscript before the 1st submission, but we should have rechecked it. We are very sorry about this. We revised the grammar and errors in the revised manuscript by professional scientific editing services (Crimson Interactive LTD and HARRISCO-ENCO Co., Ltd). We are attaching the certificate copy in the reviewer’s response for reference (Please see an attached file).

Lines 70-71: specify whether refers to mouse or human PL.

>> We apologize for this omission that may have confused the readers, including the reviewer. We have revised “PL” to specify “mouse PL” in that sentence (Line 70 and 253).

There is no significant description of the Stab 1 and Stab2 knockout mice, although references are given. A brief statement of their general non-placental phenotypes should be included as the maternal phenotype might affect placental phenotype in the current study.

>> We are very grateful to the reviewer for this valuable comment. Based on a reference (Ref#10, Schledzewski et al., 2011), Schledzewski et al studied Stab1/2 dKO mice which were generated by crossing their Stab1 KO mice and Stab2 KO mice which are different from the strategies for targeting Stab1 and Stab2 alleles compared to mouse lines which we used in this study. They reported that the survival rate of Stab1/2 dKO mice was significantly lower than that of WT, the blood clearance capacity of the liver was impaired, and the fibrosis of the kidney was significantly increased in Stab1/2 dKO mice compared to WT mice. However, they did not focus on elucidating the cause of low survival rate of Stab1/2 dKO mice. Moreover, no specific physical feature was reported in each Stab1 KO and Stab2 KO mouse line used in the present study (Ref #8 and #23). We have added a more detailed description of the non-placental phenotypes of Stab1/2 dKO mice in the Discussion section (Lines 212-219).

Results:

Line 84 and elsewhere: ‘real-time PCR and RT-PCR’ are misleading. Real-time PCR is also ‘RT-PCR’. Use ‘quantitative real-time RT-PCR’ and ‘semi-quantitative RT-PCR’ or other more precise alternatives.

>> We appreciate and agree with the reviewer’s suggestion. Accordingly, real-time PCR was revised to quantitative PCR (qPCR) in the revised manuscript.

Lines 112 -113 and reference to ‘rotten eggs’ should be deleted.

>> According to the reviewer’s suggestion, the term was deleted in the revised manuscript.

Section 2.3. e.g., lines 143-144; Fig. 3; Fig. 4 etc: It is sometimes difficult to determine whether the stated genotypes refer to embryo or mother. There needs to be a more explicit specification of the maternal and embryonic genotypes for all experiments. It is unclear which effects are due to the maternal genotype and which due to the embryonic genotype in some experiments. The Supplementary Table does not clarify this issues although the explicit statement of numbers is welcome. Precise numbers should be stated throughout the Results section including in Figure legends.

>> We thank the reviewer for raising this point. We apologize for confusing the readers, including the reviewer, regarding the animals’ genotypes in the manuscript. We have added a pedigree (Fig. 2) to clarify the maternal and embryonic genotypes, as well as placental genotypes for the present study. We have also added a pedigree to explicitly specify the mouse genotypes that were shown in a supplementary table (Supplementary Table 1 has been modified to Supplementary Fig. S4 in the revised manuscript). We hope that the added pedigree will indicate the mouse genotypes used in this study more clearly. In addition, the precise numbers of samples are now stated in the Results section or Figure legends.

Line 154: remove reference to human placental abruption / relocate to Discussion.

>> As pointed out by the reviewer, the phrase “corresponding to placental abruption in human placenta” was deleted. Instead, we have discussed this point in the Discussion section (Lines 194-199).

Discussion:

It would be good to see PL data in KO crosses, although the stated small numbers available may not have allowed this.

>> We appreciate the reviewer’s suggestion. We did not measure the PL concentration in Stab1/2 dKO compared with WT female mice, but we performed PL-1 IHC staining using paraffin sections to monitor PL expression in Stab1/2 dKO and WT placenta. PL expression was observed in the placental labyrinth zone of WT females, while no clear expression pattern was observed in the labyrinth zone of placenta from Stab1/2 dKO females. However, we could not validate the accuracy of this result. Anti-Placental lactogen antibody (sc-376436, Santa Cruz Biotechnology, Inc.), which we used, exhibited a high nonspecific background. Despite our experience performing IHC, unfortunately, we did not believe that this result was scientifically convincing. We did not present this result in the manuscript but have simply included a representative image (IHC with anti-PL-1 antibody in WT and Stab1/2 dKO placenta) in the reviewer’s response for reference (Please see an attached file).

****************************************************************

The authors want to extend our sincere thanks to the Editor and the Reviewers for the helpful comments and the time they invested in reviewing our paper. We hope that this revision will satisfy the comments and requests from the Editor and the Reviewers as well as improve the overall quality of this manuscript. We also hope that the revised manuscript is now acceptable for publication in International Journal of Molecular Sciences.

Sincerely yours,

Jung-Eun Kim, Ph.D.

Department of Molecular Medicine

Cell and Matrix Research Institute

School of Medicine, Kyungpook National University

680 Gukchaebosang-ro, Jung-gu,

Daegu, 41944, Korea

Tel: 82-53-420-4949

Fax: 82-53-426-4944

Round 2

Reviewer 1 Report

I thank the authors for their thorough reply. The authors explain the difficulties in getting viable pregnancies for this study. As I indicated in my initial report, the main issue with this study is the lack of statistics (i.e. statistical comparisons with WT) to substantiate their claims. The only placental data the authors managed to provide statistics for was the PAS-stained area (Fig. 6-C). They added data regarding maternal death, but again it is only descriptive, no statistics were provided (no P value). This is relevant because the authors are making claims without proper experimental evidence. For example, in lines 225-227 the authors stated: “Here, we present novel findings indicating that Stab1 and Stab2 are associated with deciduous hemorrhage in the placenta”. But the only data presented in the paper to backup this claim is figure 7, where an image from a knockout placenta is showed, with no statistical comparison with WT, and hence this just could have been a random finding.

Therefore, in its current format, the study is just descriptive. Don’t get me wrong, I believe this type of information should be reported, but in the right format. So my advice is to change the title, where the readers can see clearly the nature of this paper, this is, the study is NOT experimental, just a description of the reproductive issues the authors found when trying to obtain Stab1/2 dKO offspring. Perhaps the authors could use the following title: “Reproductive capacity of double knockouts for stabilin-1 and stabilin-2: a descriptive study”. Importantly, the narrative of the study should be changed. The study was not designed to test the role of stabilin-1 and -2 in placentation otherwise a control group would have been included, allowing the authors to provide statistical data in all the endpoints they reported in the paper. This should be stated in the abstract and introduction, clearly indicating that this is a descriptive study, not experimental. The discussion should change as well, to indicate that proper comparisons with WT are needed in future studies to ascertain that the placental changes observed in Stab1/2 dKO are indeed the result of lacking expression of stab1 and 2, and that gene expression of these two genes in placentas of knockouts is also essential in future studies. The authors should also indicate that future embryo transfer studies (Stab1/2 dKO blastocysts into WT surrogates) should also be used to better determine the role of Stab1/2 dKO on mammalian placentation.

Author Response

Reviewer #1

>> We appreciate and agree with the reviewer’s point. According to the reviewer’s suggestion, we have revised the title so that readers can clearly see and determine the subject of this paper by reading the title. The revised title is “Identifying stabilin-1 and stabilin-2 double knockouts in reproduction and placentation: A descriptive study”. Moreover, we have revised some sentences and have clearly indicated in the Abstract and Introduction sections (Lines 32-38 and Lines 76-83, respectively) that this is descriptive rather than experimental. We also mentioned that further studies are needed to ascertain the role of Stab1 and Stab2 in reproduction and placentation in the Discussion section (Lines 195-196, 232-234, 266-276).

Reviewer 2 Report

The authors have addressed my conncerns.

Author Response

Reviewer #2

The authors have addressed my concerns.

>> Thank you very much for your evaluation.

Round 3

Reviewer 1 Report

I thank the authors for applying the suggested changes. 

Author Response

Thank your very much for your evaluation.